# Mycotoxins from Tomato Pathogenic *Alternaria alternata* and Their Combined Cytotoxic Effects on Human Cell Lines and Male Albino Rats

**DOI:** 10.3390/jof9030282

**Published:** 2023-02-21

**Authors:** Ahmed Mahmoud Ismail, Eman Said Elshewy, Sherif Mohamed El-Ganainy, Donato Magistà, Ahlam Farouk Hamouda, Khalid A. Alhudaib, Weaam Ebrahim, Mustafa I. Almaghasla

**Affiliations:** 1Department of Arid Land Agriculture, College of Agricultural and Food Sciences, King Faisal University, P.O. Box 420, Al-Ahsa 31982, Saudi Arabia; 2Pests and Plant Diseases Unit, College of Agricultural and Food Sciences, King Faisal University, P.O. Box 420, Al-Ahsa 31982, Saudi Arabia; 3Vegetable Diseases Research Department, Plant Pathology Research Institute, Agricultural Research Center (ARC), Giza 12619, Egypt; 4Department of Soil, Plant and Food Sciences, University of Bari A. Moro, 70126 Bari, Italy; 5Institute of Sciences of Food Production (ISPA), National Research Council (CNR), 70126 Bari, Italy; 6Department of Forensic Medicine and Toxicology, Teaching Hospital, Faculty of Veterinary Medicine, Benha University, Benha 13736, Egypt; 7Department of Pharmacognosy, Faculty of Pharmacy, Mansoura University, Mansoura 35516, Egypt

**Keywords:** altenuene, alternariol, cytotoxicity, gene expression, pathogenicity, phylogeny, tenuazonic acid

## Abstract

The *Alternaria* species are considered to produce a plethora of several mycotoxins constituting a risk factor for both human and animal health. This work aimed mainly to explore the cytotoxicity of a combined mixture of altenuene (ALT), alternariol (AOH), tenuazonic acid (TeA), and altenuisol (AS) toxins produced by pathogenic *A. alternata* toward human oral epithelial cells (PCS-200-014), lung fibroblast cells (WI-38), and male albino rats. The sequencing of the multi-locus, RNA polymerase second largest subunit (*rpb2*), glyceraldehyde-3-phosphate dehydrogenase (*gapdh*), and *Alternaria* major allergen gene (*Alt a 1*) was performed to infer relationships among isolated *Alternaria* species. The phylogenetic analysis of *gapdh*, *rpb2*, and *Alt-a 1* sequence data indicated that all isolates resided in *A. alternata*. The pathogenic potentiality of *A. alternata* was investigated on tomato plants cv. super strain B under greenhouse conditions, and all isolates were pathogenic to tomato plants, with significant (*p <* 0.05) variations. The ability of *A. alternata* isolates to produce mycotoxins was also explored using high-performance liquid chromatography (HPLC). All tested isolates were able to produce at least one of the assessed mycotoxins—ALT, AOH, TeA, and AS—and ALT was reported as the dominant mycotoxin, produced by 80% of *A. alternata* isolates. The cytotoxic properties of the combined mixture of ALT, AOH, TeA, and AS at concentrations of 31.25, 62.50, 125, 250, and 500 µg/mL were assessed via the MTT assay method after exposure for 24 h versus the control. The treatment of both cell lines with combined mixtures of ALT, AOH, TeA, and AS showed a dose-dependent decrease in cell viability. The highest concentrations tested at 62.50, 125, 250, and 500 µg/mL significantly decreased cell viability and caused cell damage compared to the lowest concentration of 31.25 µg/mL and the control. The cytotoxicity and genotoxicity of the combined mixtures of ALT, AOH, TeA, and AS on male albino rats were also investigated via the gene expression of (TNF-α) and using hematological (CBC), chemical (alanine aminotransferase (ALT), aspartate aminotransferase (AST) and urea and creatinine), and histopathological analyses. A marked increase was observed in the levels of ALT, AST, urea and creatinine, TNF-α gene expression, red blood cells (RBCs), white blood cells (WBCs), hemoglobin (Hb), and packed cell volume % (PCV) after 28 days of exposure relative to the untreated control. Pathological alterations were also observed in the liver and kidney tissues of rats. Conclusively, this work provides a new understanding on the cytotoxicity and genotoxicity of mycotoxins of pathogenic *A. alternata* from tomatoes.

## 1. Introduction

*Alternaria* is a haploid fungus that belongs to the *Pleosporaceaea* family of Ascomycete fungi. The fungus is ubiquitous, with widely distributed species, and is known to cause several diseases in nearly 400 plant species; in particular, *A. alternata* infects almost 100 plant species [1]. It is well-known that in areas of heavy dew, rainfall, and high relative humidity, diseases caused by *Alternaria* are the most frequent and destructive [2]. Moreover, *Alternaria* species are responsible for low productivity and yield losses in several crops. For example, *A. brassicae* and *A. brassicicola* can cause oilseed brassica yield losses reaching 10–70% [3]. In Australia, *A. japonica* and *A. brassicae* have been reported to cause yield losses of over 58% on oilseed rape [4]. Additionally, *A. solani* is considered as a devastating disease relative to potato crops that can lead to yield losses of 5–50% [5]. Meanwhile, Adhikari et al. [6] stated that annual losses in tomato yield caused by *A. alternata* and *A. solani* have been estimated at 79%. The small-spored *Alternaria* species, e.g., *A. alternata*, *A. arborescens*, *A. tenuissima*, *A. dumosa*, and *A. interrupta*, have been reported to cause diseases in Solanaceous hosts [7,8,9]. In addition to infecting fruits and vegetables, *Alternaria* species are also well-known sources of more than 70 toxins that can lead to many human health disorders [10,11,12]. The most prevalent *Alternaria* toxins are altenuene (ALT), alternariol (AOH), alternariol monomethyl ether (AME), tenuazonic acid (TeA), and tentoxin (TEN), which have been detected in several crops and food products in reasonably measurable amounts [1,13,14]. According to the European Food Safety Authority (EFSA), *Alternaria* toxins have been considered as potentially hazardous to human health [15]. *Alternaria* toxins were assigned to “emerging” mycotoxins, for which a thorough risk assessment is mostly lacking [16]. Thus far, no regulations or guidance levels have been established for these emerging mycotoxins, but maximum levels are currently under consideration by the European Commission [17]. *Alternaria* mycotoxins can be detected by using a comprehensive range of analytical techniques such as thin-layer chromatography (TLC), high-performance liquid chromatography (HPLC), high-performance thin-layer chromatography (HPTLC), gas chromatography (GC), liquid chromatography (LC) mainly with ultraviolet (UV) detection (LC-MS), and enzyme-linked immunosorbent assay (ELISA) [18,19,20]. However, among all the above techniques, HPLC is the most extensively used for the detection of *Alternaria* toxins [18,21,22].

A molecular phylogenetic approach played a vital role in delineating phylogenetic lineages within *Alternaria* and allied genera, combining them into one large genus of the *Alternaria* complex [23]. This approach led to different systematic proposals for *Alternaria*, potentially due to the use of different molecular markers for inferring phylogenetic relationships and differentiation among taxa [24]. The genetic markers used for this purpose are large subunit ribosomal DNA (LSU), glyceraldehydes-3-phosphate dehydrogenase (*gapdh*), the internal transcribed spacer region of rDNA (ITS), mitochondrial small subunit (mtSSU), *Alternaria* major allergen gene (*Alt-a 1*), translation elongation factor 1-alpha (TEF-1), RNA polymerase second largest subunit (*rpb2*), and plasma membrane ATPase [23,25,26,27,28,29,30].

Based on the reports of EFSA (EFSA, 2016), there is a need for further research to analyze the risk of these *Alternaria* mycotoxins for public health and to fill these knowledge gaps. Therefore, one of the main goals of the current study was to investigate the potential cytotoxicity of *Alternaria* mycotoxins against human cell lines and male albino rats. This study also aimed to investigate the phylogenetic relationship among a collection of small-spored *Alternaria* based on the sequence data of *Alt a-1 gapdh* and *rpb2* gene regions. The virulence profile of *Alternaria* species in this study was also determined via artificial pathogenicity tests under greenhouse conditions.

## 2. Materials and Methods

### 2.1. Sampling and Isolation

Symptomatic leaves exhibiting small and sometimes large necrotic black lesions [10] were collected from seven Egyptian governorates—i.e., Ismailia, Sharkia, Beheira, Kafr El-Shiehk, Fayoum, Menofia, and Kaliobyia—during the 2021–2022 season. Isolation was performed as previously described [31]. Samples were washed under running water to remove surface debris and dissected into small segments of approximately 0.5 cm, followed by surface disinfection in 5% hypochlorite for 2 min and 70% ethanol for 30 s. Surface-sterilized segments (4 segments/plate) were then plated out on Potato Dextrose Agar (PDA) amended with 0.1 gm streptomycin sulfate and incubated at 25 °C. Developing hyphal tips were picked up aseptically, sub-cultured on PDA, and incubated at 25 °C. The putative isolates resembling the morphology of *Alternaria* were conserved in slant tubes on PDA medium for further studies.

### 2.2. Molecular Characterization

#### 2.2.1. DNA Extraction and PCR Amplification

Total DNA extraction from approximately 50 mg of 10-day fresh mycelium was performed using commercial DNA isolation kits following the manufacturer’s protocol. The PCR amplification of the glyceraldehyde-3-phosphate dehydrogenase (*gapdh*) region was carried out using gpd1 and gpd2 [32]. The amplification of RNA polymerase’s second largest subunit (*rpb2*) region was performed using RPB2–5F2 [33] and RPB2–7cR [34]. The *Alternaria* major allergen gene (*Alt-a 1*) region was amplified using primers alt-for and alt-rev [33]. The PCR components and amplification conditions were obtained as described by Woudenberg et al. [30]. The amplified amplicons were sequenced in both directions using PCR primers at Macrogen (South Korea) following the manufacturer’s instructions. The newly generated sequences in this study were deposited in the GenBank database (Appendix A).

#### 2.2.2. Phylogenetic Analysis

The generated sequences were manually checked and corrected where necessary and the ambiguous sequences of both 5′ and 3′ ends were excluded from the final alignment. The newly generated sequences together with other sequences obtained from GenBank were aligned using (http://mafft.cbrc.jp/alignment/server/index.html, accessed on 18 September 2022) MAFFT v. 7.0 [35]. The possible phylogenetic correlation among taxa was determined using PAUP* (Phylogenetic Analysis Using Parsimony) version 4.0b10 [36] with maximum parsimony (MP) tests. Bayesian analyses were performed by running MrBayes v3.2.7a [37] on the Cipres Science Gateway (www.phylo.org, accessed on 20 September 2022) [38] on the combined, partitioned dataset with substitution models calculated for each partition by ModelFinder on IQ-TREE multicore version 2.2.0 [39,40]. Bayesian analyses were run in duplicate with 4 Markov chain Monte Carlo (MCMC) chains, with random trees for 10,000,000 generations sampled every 1000 generations. The temperature value was lowered to 0.10, burn-in was set to 0.25, and the run was automatically stopped when the average standard deviation of split frequencies ended up below 0.01. Summaries for posterior probabilities calculation are based on a total of 3984 samples from 2 runs. Each run produced 2656 samples of which 1992 samples were included. The resulting tree was viewed and issued using FigTree version 1.4.4 (http://tree.bio.ed.ac.uk/software/figtree, accessed on 26 September 2022).

### 2.3. Pathogenicity Tests

All isolates were tested for pathogenicity as described by El Gobashy et al. [9]. Apparently healthy seedlings, 30 days old, were transplanted in plastic pots (25 cm in diameter) containing 2:1 peat moss and a sand–soil mixture disinfested with 5% formalin. The inoculum was prepared by culturing each of the tested isolates on a PDA medium at 25 °C for 15 days. Then, 10 mL of sterile distilled water was added to each plate, and colonies were carefully scraped with a sterile needle. The obtained conidial suspension was adjusted to 5 × 10^6^ spores/mL and sprayed on a leaf area. The experimental setup was maintained in three replicates per each isolate, and each replicate consisted of five plants. After inoculation, plants were covered with polyethylene bags for 48 h to maintain high-humidity conditions. After 48 h, bags were removed, and plants were kept under greenhouse conditions. These plants were regularly monitored for disease development with respect to uninoculated control leaves. Two weeks later, the disease severity was recorded using a 0–9 scale chart (0 = healthy; 1 = 1–5%; 2 = 6–10%; 3 = 11–25%; 5 = 6–50%; 7 = 51–75%; 9 ≥ 76% of the leaf area infected), as proposed by Latha et al. [41].

### 2.4. Mycotoxin Production and Extraction

Toxic metabolites of the most virulent 10 representative *A. alternata* isolates were extracted [42] with modifications. Agar plugs (5 mm) from 5 days old were grown in a 500 mL flask containing 100 mL Potato Dextrose Broth (PDB) medium and incubated for 20 days at 25 °C. Subsequently, 20-day-old cultures were filtered through Whatman No. 1 filter paper. Each filtrate was amended with an equal volume of methanol and kept at 4 °C for 24 h. After that, the mixture was shaken and evaporated at 40 °C near dryness in a rotary evaporator. The extracted filtrate was amended with an equal volume of ethyl acetate and mixed properly in a separator funnel. The aqueous layer was extracted with ethyl acetate. The ethyl acetate extract was concentrated at 44 °C and diluted in 2 mL methanol before analysis.

### 2.5. HPLC Analysis

High-performance liquid chromatography (HPLC) analyses were made using a Shimadzu HPLC apparatus equipped with a fluorescence detector and two binary gradient pumps [43]. Samples were chromatographically separated using an SCIEX AAA C18 (150 mm long × 4.6 mm, 5.0 μm particle size) column (Foster City, CA, USA). The mobile phase consisted of a mixture of MeOH: 0.1 M NaH2PO4 (2:1 *v*/*v*), adjusted to pH 3.2 and maintained with phosphoric acid. Chromatographic separation was carried out using continuous isocratic elution with 50% of eluent A and 50% of eluent B, and the flow rate was 1.0 mL/min throughout the entire separation process. The column thermostat maintained the temperature at 30 °C, and the injection volume was 25 μL. The detection and quantity of *Alternaria* mycotoxins were determined at a wavelength of 279 nm.

### 2.6. Cytotoxicity Assays

#### 2.6.1. Cell Lines

Oral epithelial cells PCS-200-014 (Primary gingival keratinocytes (PGK; ATCC^®^ PCS-200-014™) were purchased from American Type Culture Collection (ATCC; Manassas, VA, USA). Lung Fibroblast cells WI-38 were obtained from VACSERA Tissue Culture Unit, Cairo, Egypt. The cells were propagated in Dulbecco’s modified Eagle’s medium (DMEM) supplemented with 10% heat-inactivated fetal bovine serum, 1% L-glutamine, HEPES buffer, and 50 µg/mL gentamycin. All cells were maintained at 37 °C in a humidified atmosphere with 5% CO_2_ [44].

#### 2.6.2. Cell Viability Assay

The cytotoxicity of the crude extract of a combined mixture of mycotoxins ALT, AOH, TeA, and AS on PCS-200-014 and WI-38 cell lines was estimated using the MTT assay method [45,46]. Cells were seeded in a 96-well plate at a concentration 1 × 10^4^ cells per well in 100 µL of growth medium. The mycotoxin mixture of ALT, AOH, TeA, and AS at concentrations of 31.25, 62.5, 125, 250, and 500 µg/mL was added to cell cultures after 24 h of seeding. Serial two-fold dilutions of the tested mycotoxins were added to confluent cell monolayers and dispensed into 96-well, flat-bottomed microtiter plates (Falcon, NJ, USA) using a multichannel pipette. The microtiter plates were incubated at 37 °C in a humidified incubator with 5% CO_2_ for 24 h [44]. Glacial acetic acid (30%) was then added to all wells and mixed thoroughly; then, the absorbance was measured using a microplate reader (SunRise, TECAN, Inc, San Diego, CA, USA) at wavelengths of 490 and 590 nm for PCS-200-014 and WI-38 cell lines, respectively. Cell viability was calculated using the formula described by Handayani et al. [47]. The inhibitory concentration (IC_50_) value was estimated from graphic plots of the dose–response curve for each concentration using GraphPad Prism software (San Diego, CA, USA).

#### 2.6.3. Microscopic Examination

After the end of the treatment, plates were inverted to remove the medium, the wells were washed three times with 300 µL of phosphate-buffered saline (pH 7.2), and then the cells were fixed to the plate with 10% formalin for 15 min at room temperature [48]. The fixed cells were then stained with 100 µL of 0.25% crystal violet for 20 min. The stain was removed, and the plates were rinsed using deionized water to remove the excess of stain and then allowed to dry. Cellular morphology was observed using an inverted microscope (CKX41; Olympus, Tokyo, Japan) equipped with a digital microscopy camera. The cytopathic effects were microscopically observed at 100×.

### 2.7. Potential Toxicity of the Combined Mycotoxins ALT, AOH, TeA, and AS on Albino Rats

Male albino rats with mean body weight 150–160 gm were used to assess the potential toxicity of mycotoxin extract containing ALT, AOH, TeA, and AS. Rats were obtained from the animal house of the National Research Center, Giza, Egypt. They were kept in the animal house in the Faculty of Veterinary Medicine, Benha University, at 21–22 °C and at 12 h light/dark intervals and were allowed free access water and standard food pellets throughout the experimental period (28 days). Rats acclimatized to the laboratory’s conditions for one week before starting the experiment. The experimental protocol was developed in accordance with ethical guidelines (ethical approval BUFVTM 19-10-22). Rats were randomly divided into six groups; three groups (three rats each) were caged individually and were administered a mixture of mycotoxin extract containing ALT, AOH, TeA, and AS daily for 28 days by oral gavage [49]. The remaining three groups (three rats each) were administered distilled water (negative control). At the end of the experiment, blood samples were collected by cardiac puncture in vacutainers with and without EDTA for either hematological analysis or serum biochemical analysis. Rats were weighed on the 1st day of the study and at the end of the experiment.

### 2.8. Gene Expression Levels of Apoptosis-Associated Gene (TNF-α) by RT-PCR

At the end of the experiment, samples from liver tissues were excised from the sacrificed animals of each group and preserved at −80 °C. The expressions of mRNAs for Tumor Necrosis Factor-α (TNF-α) gene were quantified using RT-PCR [50]. The total RNA was extracted from liver tissues (approximately 40 mg) using the RNAEasy kit (Qiagen, Germany), following the manufacturer’s protocol. RT-PCR reaction and amplification conditions were obtained as described by Khan et al. [51]. Glyceraldehydes-3-phosphate dehydrogenase (*gapdh*) was used as a housekeeping gene for normalizing expression data [52].

### 2.9. Hematological Analysis

Whole-blood samples were collected and subjected to CBC tests; hemoglobin concentration (Hb), packed cell volume % (PCV%), and white blood cell (WBC) and red blood cell (RBC) counts were estimated using an electronic cell counter (VetScan HM5 Hematology system, Abaxis, Inc., Union City, CA, USA).

### 2.10. Serum Biochemical Analysis

Harvested sera were used for the determination of the activities of aspartate transaminase (AST) and alanine transaminase (ALT) [53], the concentration of urea [54], and creatinine levels [55].

### 2.11. Histopathological Analysis

At the end of the experiment, autopsy samples were taken from the liver, kidney, and spleen of treated and untreated (control) rats and fixed in 10% formal saline for 24 h. Washing was performed using tap water, and then serial dilutions of alcohol (methyl, ethyl, and absolute ethyl) were used for dehydration. Specimens were cleared in xylene and embedded in paraffin at 56 °C in a hot air oven for 24 h. Paraffin bee wax tissue blocks were prepared for sectioning at 4-micron thickness using a rotary LEITZ microtome. The obtained tissue sections were collected on glass slides, deparaffinized, and stained by hematoxylin and eosin stain [56] for examination using a light electric microscope.

### 2.12. Statistical Analysis

All data were subjected to the statistical analysis of variance [57] and presented as the mean ± standard deviation (SD). All tests were performed in SPSS 16.0 statistical software (SPSS Inc., Chicago, IL, USA), and means were separated by Fisher’s protected least significant difference (LSD).

## 3. Results

### 3.1. Sampling and Isolation

In total, the 20 isolates represented different geographical areas, and different colony morphologies and tomato cultivars were used for further studies. They all belonged to the genus *Alternaria*, and the identification was further confirmed based on the multi-locus analysis of *gapdh*, *rpb2*, and *Alt-a 1* genes.

### 3.2. Phylogenetic Analysis

The generated PCR products are approximately 616 bps for the *gapdh* region, 514 bps for the *Alt-a 1* region, and 500 bps for the *rpb2* region. The individual BLASTn search results of sequences *gapdh*, *rpb2*, and *Alt-a1* indicated that all isolates resided in *A. alternata*. The final sequence alignment of the combined dataset comprising 46 taxa included 20 isolates obtained in this study and 26 *Alternaria* reference sequences retrieved from Gen Bank, including *A. alternantherae* CBS 124392 as the out-group taxon. The combined sequences yielded a dataset consisting of 1515 characters; 149 characters were parsimony-uninformative, 141 were parsimony-informative, and 886 (proportion = 0.839) were constant. A heuristic search with the random addition of taxa (1000 replicates) resulted in a phylogenetic tree (TL = 370 steps; CI = 0.854; RI = 0.918; RC = 0.784; HI = 0.145), and one of the most parsimonious trees is presented in Figure 1.

### 3.3. Pathogenicity Tests

The pathogenic potentiality of the 20 isolates of *A. alternata* was evaluated against tomato cv. super strain B to prove Koch’s postulates. The first appearance of the disease’s symptoms was observed after 8 days of incubation, and the symptoms comprised brown necrotic lesions on leaves with irregular margins (Figure 2A,B). According to the statistical analysis of variance, there were significant (*p <* 0.05) differences among isolates in terms of disease severity (DS), and isolate ALT2224 (mean ± standard deviation (SD) = 33.44 ± 1.36) was the most aggressive relative to all isolates (Figure 2D). However, isolates ALT2258 (mean ± SD = 28.83 ± 0.67), ALT2261 (mean ± SD = 28.47 ± 1.67), and ALT2215 (mean ± SD = 30.42 ± 1.55) also exhibited high DS values but still were not significantly (*p <* 0.05) different among each other (Figure 2D). By contrast, isolates ALT2246 and ALT2213 showed the lowest DS on tomato plants (Figure 2D). No lesions developed on the tomato plants sprayed with sterile water (Figure 2C). The tested *Alternaria* isolates were successfully re-isolated from inoculated tomato plants, thereby fulfilling Koch’s postulates.

### 3.4. HPLC Analysis

The HPLC analysis of ten *A. alternata* isolates showed that all tested isolates were able to produce at least one of the screened mycotoxins. The chromatogram peaks confirmed the presence of ALT, AOH, TeA, and AS toxins in the tested isolates (Appendix A). The results clearly indicate that the highest value of ALT (10.12 µg/mL) was produced by isolate ALT2210, followed by ALT2261 (7.26 µg/mL), ALT2215 (7.11 µg/mL), and ALT2257 (6.81 µg/mL) (Figure 3 and Figure 4). However, the other isolates produced ALT at lower levels (1.12–5.11 µg/mL). ALT was not produced by the ALT2254 isolate. On the other hand, the highest value of AOH was produced by isolate ALT2257 (18.39 µg/mL), and the remaining isolates produced AOH at lower levels ranging from 1.24 to 8.56 µg/mL. AOH was not detected by isolates ALT2210, ALT2254, and ALT2232. The results also revealed that the highest value of TeA was produced by ALT2254 (11.94 µg/mL). The rest of the isolates produced TeA with values ranging from 2.19 up to 8.02 μg/mL. TeA was not detected in the following isolates: ALT2265, ALT2244, and ALT2257 (Figure 3). The mycotoxin AS was produced by only two isolates, ALT 2265 and ALT2244, with values of 1.55 and 10.29 μg/mL, respectively (Figure 3).

### 3.5. Cytotoxicity Assay

The results of the MTT assay revealed that the viability of oral epithelial cells PCS-200-014 (Figure 4A) and lung fibroblast WI-38 (Figure 4B) cell lines decreased depending on the concentration of *Alternaria* mycotoxins ALT, AOH, TeA, and AS. Notably, the exposure of PCS-200-014 cells for the 24 h treatment with mycotoxins at the lowest concentrations (31.25 and 62.5 µg/mL) did not induce any cytotoxic effects compared to the control (Figure 4A,B). By contrast, the concentrations of 31.25 and 62.5 µg/mL caused a slight decrease in the viability of WI-38 cells, reaching 85.36% and 52%, with inhibition reaching 14.64 and 47.96%, respectively (Figure 4A,B). In contrast, the highest concentrations of 250 and 500 µg/mL induced stronger cytotoxic effects that reduced the viability to 39.41 and 11.38% for PCS-200-014 cells and 20.87 and 6.79% for WI-38 cells, respectively. Furthermore, a moderate reduction in cell viability of 80.71% was observed in PCS-200-014 cells after the exposure to the 24 h treatments, and the reduction amounted to a concentration of 125 µg/mL (Figure 4A,B). However, this concentration induced a significant decrease in the viability of WI-38 cells to 36.56% after 24 h of exposure. We conclude that the current results confirmed that ALT, AOH, TeA, and AS were cytotoxic relative to mammalian cells in a dose-dependent manner.

### 3.6. Microscopic Examination

Morphological abnormalities were observed in both treated WI-38 and PCS-200-014 (Figure 5) cells exposed to combined mixtures of *Alternaria* mycotoxins (ALT, AOH, TeA, and AS) at concentrations of 125, 250, and 500 µg/mL, respectively. These abnormalities appeared as follows: the deterioration of cells and cell walls. The rounding of cells was also observed in the same culture. Moreover, cytoplasmic shrinkage and the formation of blebs on cell surfaces were observed and resulted in the generation of apoptotic bodies, which indicated cell death compared with untreated control cells. When the concentration of the mycotoxin elevated to 500 µg/mL, PCS-200-014 cells were scarcely observed (Figure 5). These results indicated that PCS-200-014 cells were more sensitive to *Alternaria* mycotoxins than WI-38 cells.

### 3.7. Potential Toxicity of the Combined Mycotoxins ALT, AOH, TeA, and AS on Male Albino Rats

#### 3.7.1. Gene Expressions of mRNAs for TNF-α and Body Weight

A significant inflammatory response was observed in the release of TNF-α in the rat liver after 24 h of exposure to combined *Alternaria* mycotoxins (ALT, AOH, TeA, and AS) (Figure 6A). The results exhibited that the expression of the proinflammatory TNF-α gene was significantly (*p* < 0.01) upregulated in treated rats, with a value 1.328, compared to the untreated control, exhibiting a value of 0.487. Furthermore, no significant decrease in the body weight gain of treated rats compared with their initial body weight and that of the control group rats (Figure 6B) was observed.

#### 3.7.2. Hematological Examinations

Hematological analyses revealed that treated rats exhibited a slight decrease of 3.5 (×10^6^/µL) in RBC count, but this decrease was not significantly (*p* < 0.01) different when compared to untreated control rats’ 3.9 (×10^6^/µL) (Figure 6C). However, a significant (p < 0.01) decrease of 9.4 (×10^3^/µL) in the WBC count was observed when compared to the untreated control, exhibiting 19.67 (×10^3^/µL) (Figure 6D). Furthermore, no significant (*p* < 0.05) increase in the Hb and PCV % was observed after 28 days in treated rats relative to the untreated control (Figure 6E,F).

#### 3.7.3. Liver and Kidney Functions

The enzymatic activity of ALT and AST as an indicator of liver function was assessed in rats treated with combined *Alternaria* mycotoxins (ALT, AOH, TeA, and AS) (Figure 6G). Rats treated with mycotoxins exhibited a significant (*p* < 0.01) increase in ALT and AST levels, with values reaching 90.10 and 353.33 U/L, respectively, when compared to the untreated control (Figure 6G). Moreover, there were significant (*p* < 0.01) increases in serum urea and creatinine levels in treated rats, which were recorded at 41.87 and 1.10 mg/dL, respectively (Figure 6H). However, this increase did not indicate kidney failure.

#### 3.7.4. Histopathological Examination

The microscopic examination of hepatic tissues showed no morphological changes in the hepatic parenchyma of the liver of untreated rats (Figure 7A). In contrast, the hepatic parenchyma of rats treated with mycotoxins exhibited focal circumscribed round granuloma-like formations consisting of mononuclear leucocytes and eosinophils cells surrounded by fine fibroblastic cell proliferation (Figure 7B–D). The livers of these rats also showed pathological changes in the portal areas, including inflammatory cell infiltrations and portal vein dilatations (Figure 7E,F). Meanwhile, no histopathological alteration of the glomeruli and tubules at the cortex of the kidney of untreated control rats was recorded (Figure 8A). However, treated rats showed pathological alterations in the kidney’s tissues, including perivascular edema and inflammatory cell infiltration surrounding the dilated blood vessels at the cortex (Figure 8B,C). Furthermore, histopathological alterations, such as the focal hemorrhage of extravagated red blood cells in between the tubules (Figure 8D,E) and congestion in glomerular tufts and intertubular blood vessels (Figure 8F), were clearly observed in the kidneys of these rats. Moreover, rats treated with *Alternaria* toxins showed lymphoid depletion in the white pulps (Figure 9B), while the spleen of the untreated control showed no histopathological alterations in the lymphoid cells at the white and red pulps, as well as the sinusoids (Figure 9A).

## 4. Discussion

*Alternaria* species are known as major plant pathogens and have been frequently reported in solanaceous crops, particularly tomatoes and potatoes. The dominant isolation of *A. alternata* from tomatoes in the current study corresponded to previous studies that demonstrated the cosmopolitan distribution of *A. alternata* as a causal agent of leaf spot and blight on tomatoes [9,13,58], potatoes [59,60], and other hosts [61] in Egypt. Our results revealed that *A. alternata* isolates showed a marked morphological variability (data not shown). This variability in morphological characteristics could be attributed to the variation in temperature and their geographical origins in Egypt [1,13,62,63]. Thus, we were not able to characterize *Alternaria* isolates based on their morphological characteristics, as they may not reflect their accurate identity. Recent molecular studies relied on several multi-locus genes, i.e., ITS, LSU, SSU, tef1-*a*, *Alt-a1*, *gapdh*, and *rpb2*, for the characterization of *Alternaria* species, including *Alternaria* sect. *Alternaria* [64,65]. Although ITS, LSU, SSU, and tef1-*a* could not be used to delineate some species in sect. *Alternaria*, the remaining gene regions of *Alt-a1*, *gapdh*, and *rpb2* reflected six novel species belonging to the genus *Alternaria* sect. *Alternaria* [64]. Therefore, in the present study, we relied on the data of the multi-locus phylogenetic analysis of three gene regions—*Alt-a1*, *gapdh*, and *rpb2*—that were clearly sufficient for separating *A. alternata* from other species.

Statistical analyses of data indicated that the virulence of the tested isolates of *A. alternata* could be clearly differentiated during the experimental period elapsed from 8 to the maximum of 30 days. Despite morphological and genetic variabilities, all tested isolates of *A. alternata* were pathogenic and induced typical symptoms of leaf spots on tomato plants. Our results revealed that the highly virulent isolates were ALT 2224, ALT2215, and ALT2258, which produced three types of mycotoxins, ALT, AOH, and TeA, at high concentrations. Therefore, we can suggest that these mycotoxins were related to fungal pathogenicity, indicating that they were phytotoxic and played a vital role in pathogenicity [31,59,66,67].

The potential capacity of *A. alternata* isolates to produce mycotoxins has also been evaluated. Almost all strains were able to produce at least one of three assessed mycotoxins (ALT, AOH, TeA, and AS)—sometimes at a high level. However, the mycotoxin quantities produced were variable among isolates for all mycotoxins. Altenuene (ALT) was reported as the dominantly produced mycotoxin by 80% of *A. alternata* isolates, with a dramatic variability in production levels (up to 10.12 µg/mL). By contrast, recent studies reported that TeA is the most abundant mycotoxin produced by *A. alternata* when compared to ALT and AOH [9,31,68,69]. In addition, the co-production of AOH and TeA should be taken into account since 70% of *A. alternata* isolates had the capacity to produce such mycotoxins. This finding is of concern, as these mycotoxins have been associated with several diseases in humans and animals [68,70,71].

Numerous studies explored the cytotoxic properties of different mixtures of mycotoxins, and such combinations often exhibit different levels of cytotoxicity compared to individually applied mycotoxins, with stronger cytotoxicity [46,72,73,74]. There are few studies investigating the combinatory effects of *Alternaria* mycotoxins [72]. Of these, Bensassi et al. [75] investigated the cytotoxicity of *Alternaria* mycotoxins AOH and AME and found that both toxins reduced cell viability by about 30% after 24 h of exposure when tested individually, while they reduced viability by about 50% when tested in combination. Additionally, Fernández-Blanco and co-authors reported synergistic cytotoxic effects in Caco-2 cells after 24 h of exposure to AOH and AME [76]. The obtained results of the combined mixture of ALT, AOH, TeA, and AS exhibited cytotoxic effects in a dose-dependent manner on WI-38 and PCS-200-014 cell lines. A similar trend has been recently reported, since the combined mixture of AOH, AME, and TeA revealed a significant decrease in the viability of human intestinal epithelial (Caco-2) cells [77]. The current study corroborated these findings, since the highest concentrations, 250 and 500 µg/mL, of ALT, AOH, TeA, and AS induced a stronger decrease in the viability of WI-38 and PCS-200-014 cells when compared to the lowest one.

Morphological alterations in WI-38 and PCS-200-014 cells, such as irregular growth patterns and elongated arms, were observed after exposure to ALT, AOH, TeA, and AS for 24 h. These results are in line with those of Groestlinger et al. [50], who reported that the binary mixture of *Alternaria* toxins AOH and altertoxin II (ATX-II) significantly altered the architecture of colon epithelial cells (HCEC-1CT) after 24 h of incubation. Similarly, ATX-II was also able to predominantly alter membrane functionality in intestinal epithelial cells (HCECs), thus hampering crucial functions for cellular motility [78]. Moreover, the binary mixture of AOH and AME could induce multiple morphological changes in the human bronchial epithelial cell line (BEAS-2B) after exposure for 24 h [79].

The expression mRNA of TNF-α in the liver tissues of albino rats in response to exposure to a mixture of *A. alternata* mycotoxins—ALT, AOH, TeA, and AS—was investigated by RT-PCR. Corresponding well with the recent literature [80], an inflammatory response was provoked by mycotoxins. In detail, TNF-α mRNA levels were significantly upregulated by *Alternaria* mycotoxins, indicating the apoptotic pathway. Additionally, the obtained results corresponded to those previously recorded [81,82], in which AOH enhanced the mRNA level of TNFα, and it was also found to induce DNA damage. Interestingly, rather than *Alternaria* mycotoxins, RT-PCR revealed that the oral gavage of rats with *Fusarium* mycotoxin deoxynivalenol (DON) rapidly (1 h) induced TNF-α mRNA levels and interleukin-6 mRNA and protein expression in several organs and plasma [83]. Moreover, rats exposed to Satratoxin G (SG), a mycotoxin derived from the fungus *Stachybotrys chartarum*, exhibited elevated mRNA expression for proinflammatory cytokines and the level of TNF-α [84].

Weight gain in mice has been used as the indicator for establishing the tolerable daily intake of mycotoxins [85,86]. The results presented herein indicate that *A. alternata* mycotoxins slightly decreased the body weight gain of treated rats (205.67 g) compared with their initial body weight (153.3 g) and that of control group rats (214.67 g) after 28 days of oral gavage. This finding reflected the cumulative alterations of the mycotoxin treatment, which is in agreement with observations made by several investigators [85,86,87,88,89].

The complete blood count test is one of the most commonly performed blood tests, as it can tell us more about the health status. In the present study, a minor decrease in the level of RBC and Hb blood parameters and a significant (*p* < 0.01) decrease in WBCs were observed. This decrease in WBCs indicates infection, damage to tissue, leukemia, or inflammatory disease. These results are in agreement with those reported in [85,87,90], outlining that *Alternaria* mycotoxins might compromise the immune system. ALT and AST are important enzymes located in the hepatocytes of the liver cell, reflecting its function. The significant (*p* < 0.01) increase in the activity level of these enzymes in the serum of the treated rats is a result of damages caused to the liver cell by exposure to mycotoxins, which is complimentary to the findings of earlier researchers [85,87]. Consistent with earlier reports [85,87,91], we observed a significant (*p* < 0.01) increase in the levels of creatinine and urea in the serum of mycotoxin-treated rats after 28 days of oral gavage.

Hepatic, renal, and spleen histopathological alterations were also observed in response to *A. alternata* mycotoxin exposure, such as inflammatory infiltrations, portal vein dilatations, focal hemorrhage of extravagated red blood cells in between the tubules, and congestion in the glomerular tufts and intertubular blood vessels. Comparable results were also reported by many researchers, exploring the potentiality of *Alternaria* mycotoxins in inducing cumulative alterations with respect to organ damage after 28 days [49,85]. Correspondingly, an earlier study stated that the crude extract of *Alternaria* spp. and other fungi could induce lesions on the liver and spleen of Swiss albino rats after 7 days of administration [92]. Moreover, a very recent study [93] also demonstrated significant histopathological changes in the vital organs of mice administered the highest dose of 400 mg/kg of AOH; however, the rats administered with doses of 50, 100, and 200 mg/kg did not exhibit any histopathological alternations in their vital organs. On the other hand, [94] confirmed that male Sprague Dawley rats treated with altertoxin I (ATX-I) displayed histopathological damage in the liver, kidney, and spleen, even in a relatively low dose range, without significant effects on hematological and serum biochemical parameters. Unexpectedly, male albino Wistar rats administered AME at a daily dose of 10 mg kg^1^ for 7 weeks showed normal liver histology, indicating the nontoxic nature of AME [95].

## 5. Conclusions

In conclusion, this work provides a new understanding of the cytotoxicity and genotoxicity of combined mycotoxins ALT, AOH, TeA, and AS of pathogenic *A. alternata*. The results showed that a single isolate of *A. alternata* could produce more than one mycotoxin, making it important for assessing the cytotoxic effects of different combinations of mycotoxins. Based on our findings, such combinations of mycotoxins exhibited stronger cytotoxicity when compared to the individual effects that were discussed in previous studies. As presented in this study, toxins produced by *A. alternata* were investigated for their potential toxicity against oral epithelial PCS-200-014 and lung fibroblast WI-38 human cells, as well as rats. A selection of cell lines for cytotoxicity studies is important because different cell lines react differently to certain mycotoxins. Therefore, we used two different cell lines that reacted differently in response to different doses of mycotoxins, with PCS-200-014 being the most strongly affected. Mycotoxins exhibited cytotoxic properties in a dose-dependent manner. Cytotoxicity results evidenced the presence of morphological changes on the tested cell lines and also revealed hepatic, renal, and spleen histopathological alterations in treated rats. As a future prospect, the data presented here may provide a basis for designing further studies for deepening the knowledge about unidentified compounds of *Alternaria* mycotoxins and their associated cytotoxicity and for assessing their risks by taking into account the actual role of chemical mixtures.

## Figures and Tables

**Figure 1 jof-09-00282-f001:**
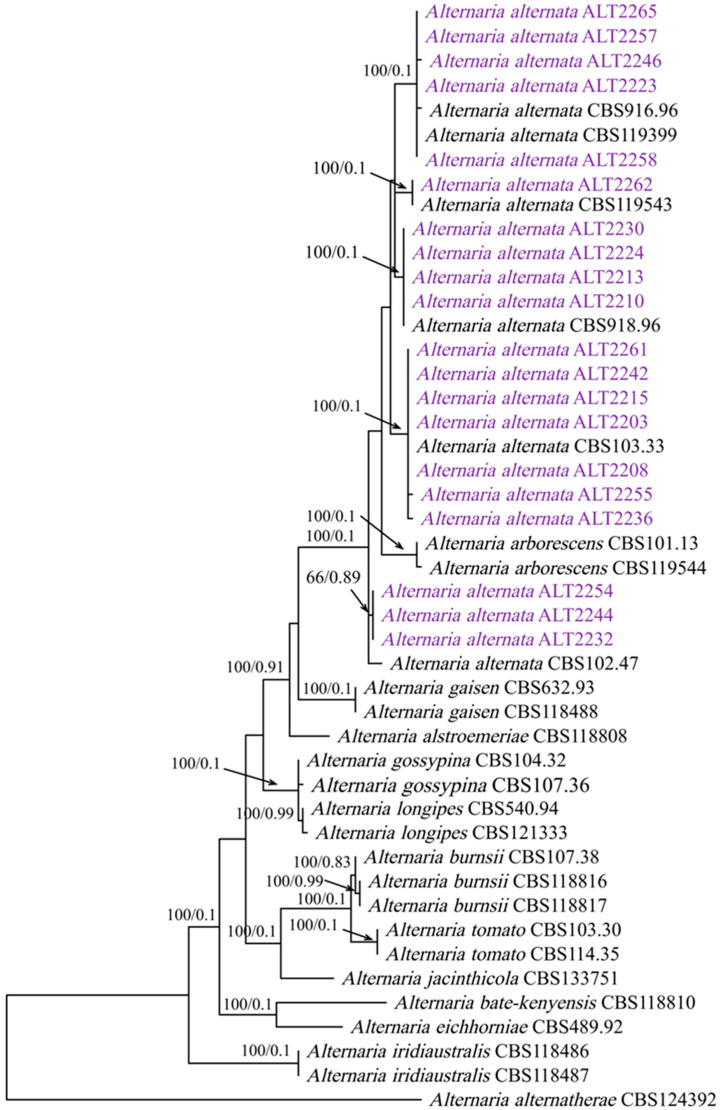
Phylogenetic tree generated by the MP-based analysis of the combined *Alt-a1*, *gapdh*, and *rpb2* DNA sequence dataset. Branches are shown on nodes with bootstrap values (BS %) and Bayesian posterior probabilities (PP). The tree is rooted to *Alternaria alternantherae* CBS 124392.

**Figure 2 jof-09-00282-f002:**
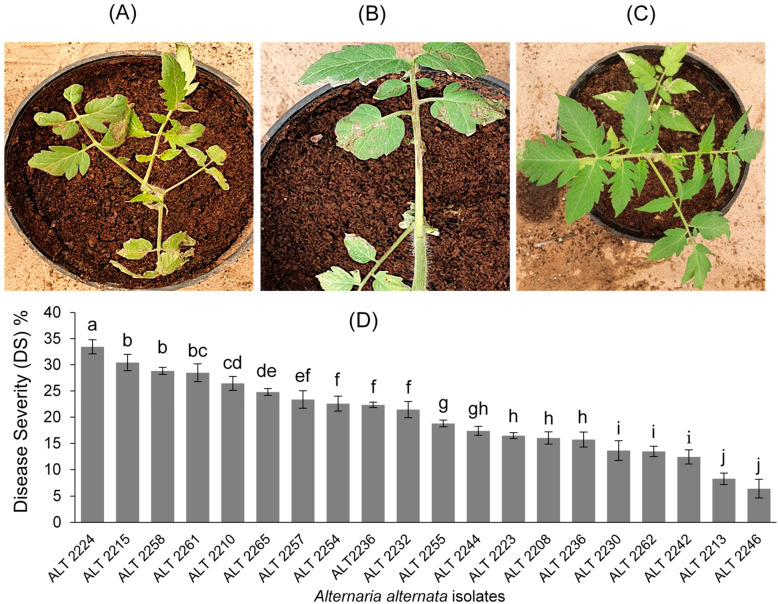
Disease severity (DS) (*y*-axis) of *A. alternata* isolates (*x*-axis) on tomato cv. super strain B over 30 days post-inoculation (**A**–**D**). Results are expressed in the mean of three replicates (*n =* 5 plants each), and vertical bars represent ± standard deviation (SD). Means having the same letters are not significantly different from each other according to the least significant difference (LSD) test (*p* < 0.05).

**Figure 3 jof-09-00282-f003:**
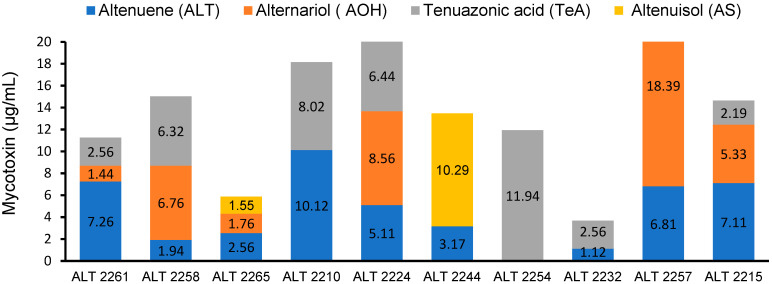
Graphical representation of the HPLC analysis results of the screened mycotoxins altenuene (ALT), alternariol (AOH), tenuazonic acid (TeA), and altenuisol (AS) and their varying concentrations from pathogenic isolates of *A. altrnata*.

**Figure 4 jof-09-00282-f004:**
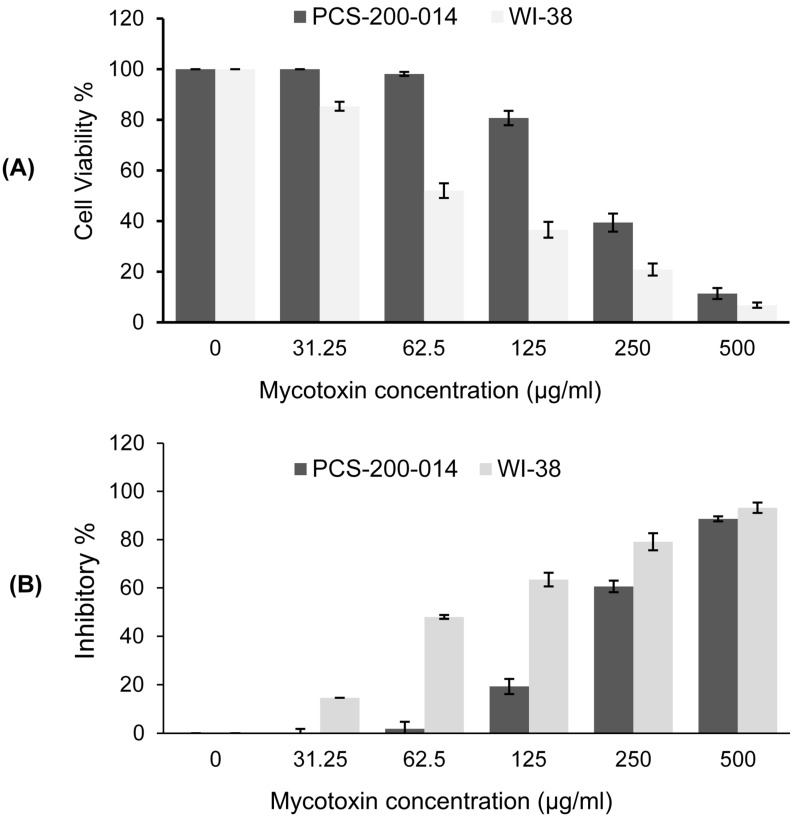
Cell viability % (**A**) and inhibitory % (**B**) of cells showing the dose–response inhibitory effect of the crude extract of combined *Alternaria* mycotoxins (ALT, AOH, TeA, and AS) against oral epithelial human cell PCS-200-014 and lung fibroblast WI-38 subjected to treatments for 24 h.

**Figure 5 jof-09-00282-f005:**
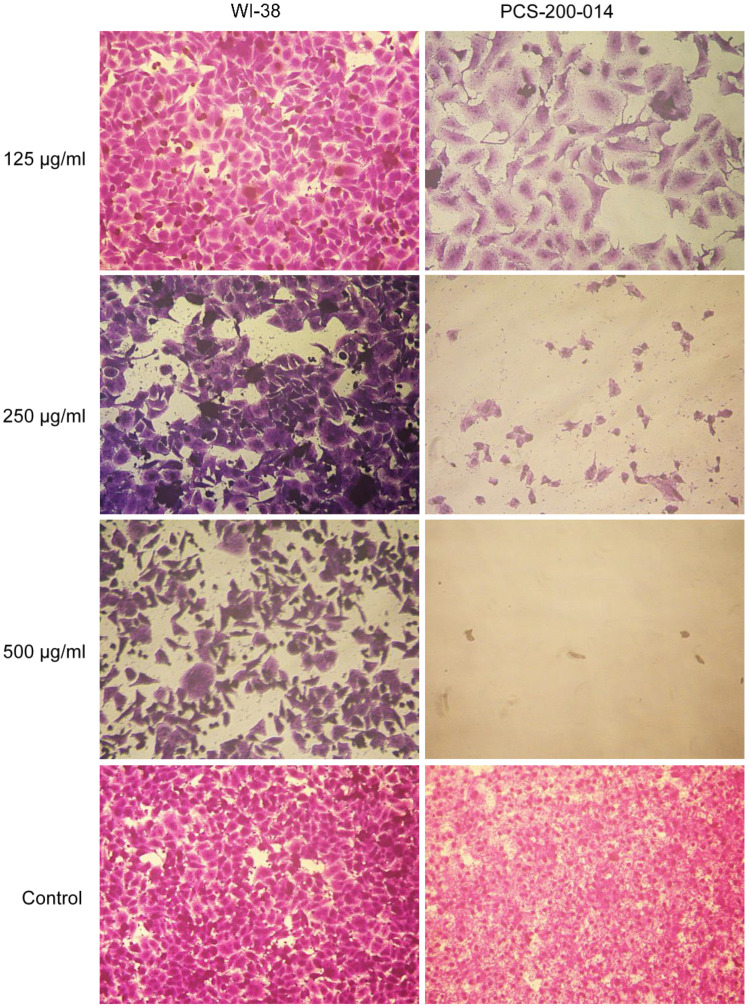
Morphological changes in WI-38 and PCS-200-014 cells after exposure to a crude extract of combined *Alternaria* mycotoxins (ALT, AOH, TeA, and AS) at concentrations of 125, 250, and 500 µg/mL for 24 h.

**Figure 6 jof-09-00282-f006:**
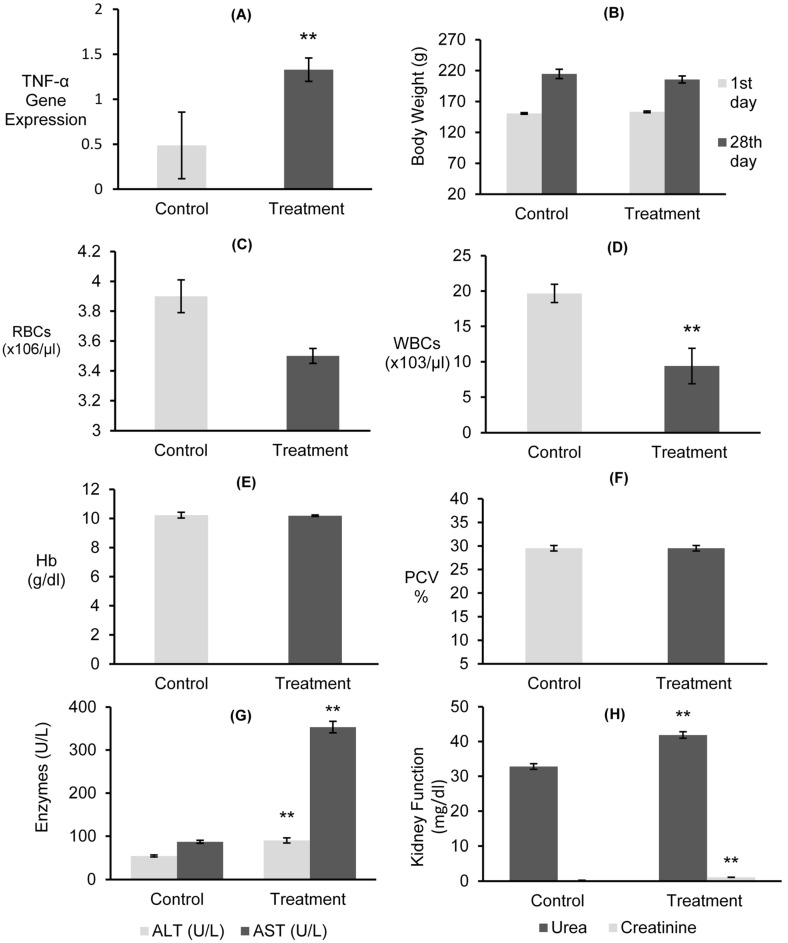
Toxic effects of the crude extract of combined *Alternaria* mycotoxins (ALT, AOH, TeA, and AS) on TNF-α gene expression (**A**), body weight (**B**), red blood cells (RBCs) (**C**), white blood cells (WBCs) (**D**), hemoglobin (Hb) (**E**), packed cell volume (PCV) % (**F**), liver enzymatic activity of alanine aminotransferase (ALT) and aspartate aminotransferase (AST) (**G**), and kidney functions (**H**) of rats compared to the control. Data are expressed as mean ± SD. Means are ** highly significant at *p* < 0.01 according to the LSD test.

**Figure 7 jof-09-00282-f007:**
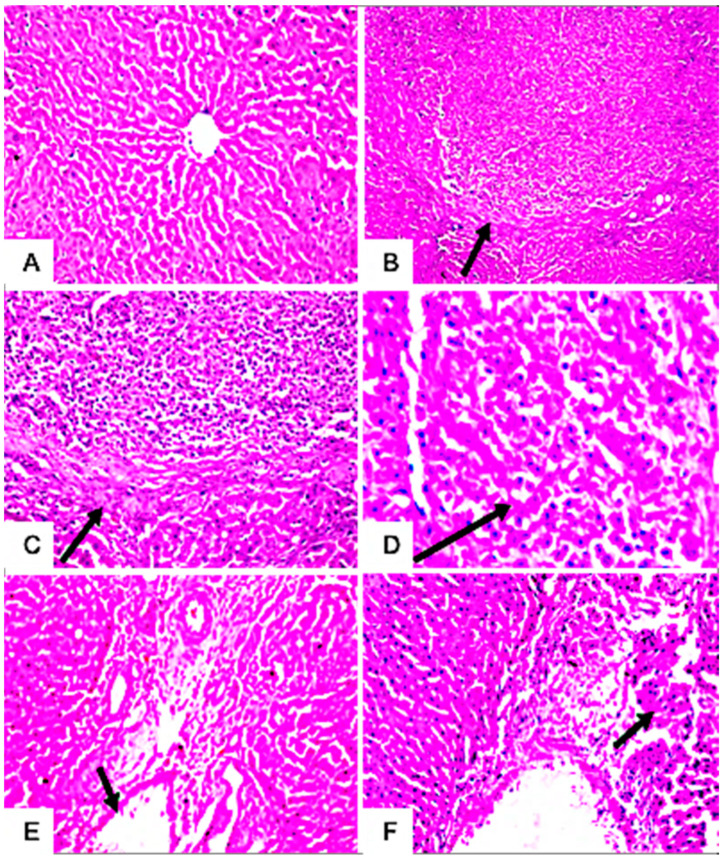
Histological changes in the liver tissues of male albino rats treated with the combined *Alternaria* mycotoxins (ALT, AOH, TeA, and AS); no histopathological alteration and the normal histological structure of the central vein and surrounding hepatocytes in the lobules of the parenchyma (**A**); focal circumscribed round granuloma-like formation consisting of mononuclear leucocytes and eosinophils cells surrounded by fine fibroblastic cell proliferation (black arrow) (**B**–**D**); the portal area showed dilations in the portal vein and inflammatory cell infiltration (black arrow) (**E**,**F**).

**Figure 8 jof-09-00282-f008:**
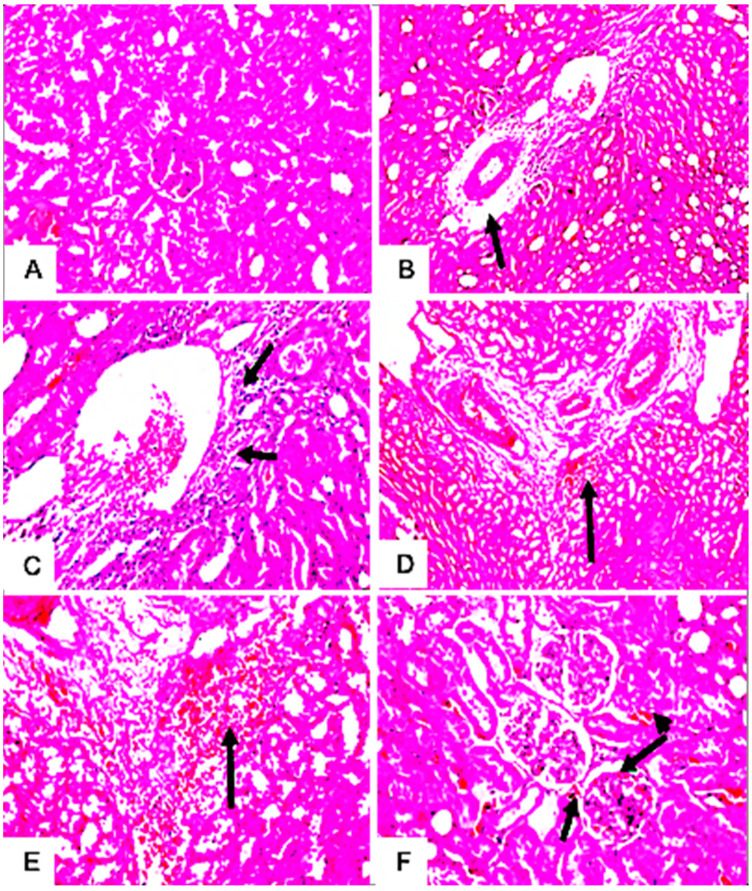
Histopathological changes in the kidney tissues of male albino rats treated with the combined *Alternaria* mycotoxins (ALT, AOH, TeA, and AS); no histopathological alteration in the glomeruli and tubules at the cortex were recorded in untreated and treated rats (control) (**A**); perivascular edema and inflammatory cell infiltration were detected surrounding the dilated blood vessels at the cortex (black arrow) (**B**,**C**); focal hemorrhage of extravagated red blood cells were observed in between the tubules (black arrow) (**D**,**E**); a congestion in the glomerular tufts and intertubular blood vessels (black arrow) was also detected (**F**).

**Figure 9 jof-09-00282-f009:**
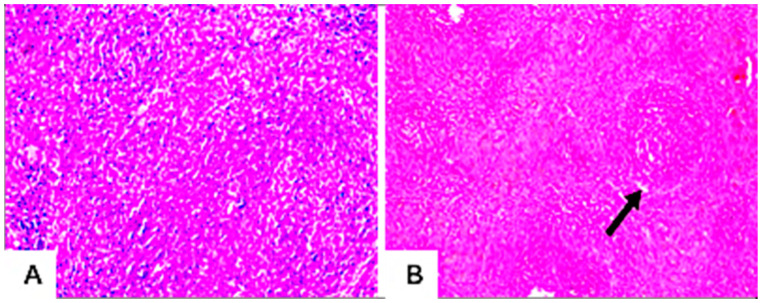
Histological examination of spleen tissues of albino rats; no histopathological alteration was observed in the histological structure of the white and red pulps, as well as sinusoids in untreated (control) rats (**A**); lymphoid depletion was detected in white pulps (black arrow) (**B**).

## Data Availability

All the data related to this study are mentioned in the manuscript.

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
