# Peer review of "Mycotoxins from Tomato Pathogenic Alternaria alternata and Their Combined Cytotoxic Effects on Human Cell Lines and Male Albino Rats"

_jof, 2023, doi:10.3390/jof9030282_

Round 1

Reviewer 1 Report

Hello dear,

The important points related to the manuscript entitled “ Mycotoxins from Tomato Pathogenic Alternaria alternata and 2 their Combined Cytotoxic Effects on Human Cell Lines and 3 Male Albino Rats” which is also mentioned in the manuscript, are as follows;

 General comment

What is the innovation of this study?

 ABSTRACT

L 50- Would you like to emphasize the locality of the study?

L 51- “HPLC”Is it key?

 INTRODUCTION

L 60- Please bring more up-to-date sentences and statistics to support and introduce the topic. The 2012 reference is not sufficient and appropriate

 MATERIALS AND METHODS

2.1. Sampling and Isolation : Please provide appropriate references in each section

 L 99-100- Why is sampling limited to Egypt?

 L 99- “small to large necrotic black lesions” What is meant by small and large? By mentioning valid reference

Was it not possible to induce contamination in the laboratory?

2.13. Statistical Analysis OR 2.12? Check the order of the numbers

 RESULT

Figure 3 ; Please add . after number

Figure .7; Please correct the . place here and in whole manuscript

 DISCUSSION

In the discussion section, the authors have paid attention to the results very superficially. It is necessary to give in-depth analysis along with the relevant mechanism in this section. There is a serious need to rewrite the discussion to fit the mechanisms

L 609- Despite the variety of data in the histopathology section, it has been reduced to a small paragraph?

 CONCLUSIONS

Much of the conclusion seems to be focused on axioms. Lack of innovation or defects in the correct narration of study results. Need for serious revision

L 620-622: “These outcomes, enhance our 620 knowledge that mycotoxins pose a vital threat to human and animal health and can prominently affect their performances and production.” Such a conclusion is not appropriate at this level

 Best regards,

Author Response

Dear Distinguished Professor,

Thank you very much for your interest in our manuscript and for meaningful comments that will certainly add to our work. We followed all the suggestions and recommendations illustrated by the reviewer’s comments.

All revisions made to the manuscript were marked up using the “Track Changes” function to be easily viewed by the editors and reviewers.

Reviewer 2 Report

Manuscript ID: jof-2194635

Title: “Mycotoxins from Tomato Pathogenic Alternaria alternata and their Combined Cytotoxic Effects on Human Cell Lines and Male Albino Rats”

Reviewer’s comment

General comments:

 The authors have addressed interesting and current topic about the formation of mycotoxins from A. alternata that are dangerous to human health. The combined chemical, biological and gene expression approach used to better assess the cytotoxic effects of single mycotoxin or the mixture of them, is very attractive.

The manuscript is written in easy way and is well organized.
However, the paper in my opinion lacks real conclusions and the reader's expectations after the presentation of the paper (title and the last sentence of abstract) are disregarded.

It would have been very interesting to have an overall evaluation of the results, because as they are reported they are unrelated to each other. The paper loses value and seems to be multiple works summarized in one text. For example, the results of cytotoxicity tests are not at all discussed or related to other tests, the question that arises is whether or not you think they are necessary to assess the toxins action or why they were done in this work.

I suggest to the authors reviewing the conclusions with a view not only to reporting what was found, but to speculate possible application of such a complex assessment approach.

I would suggest some issues should be addressed and these are outlined below.

 Introduction:

The authors explain the problem they deal with well, but when checking the bibliography reported in the text, I sometimes find it difficult to believe that the references given are correct.

(for examples: i) line 64 references 7-9; ii) line 72 reference 14: are they slides from a presentation? iii) line 89 missing the reference)

Please, I ask you to review the references and replace them with new ones more relevant to the sentence when possible.

 Material and methods

I suggest you to put the table 1 supplementary materials.

 Results

Figure 2.  the graph on Disease Severity (DS) is not immediately comprehensible. I would like to make some suggestions:

- can you put the A. alternaria isolates in ascending order because they are easier to spot (or in order to SD value)

- can you better explain the significance of the difference in severity disease among A. alternaria isolates?  A sentence about it could be added in the text as well as in the note on the figure.

 Figure 3. This figure is certainly interesting, but also needs more clarity: you can clearly see which isolates produce toxins, but in my opinion, it is repetitive compared to Figure 4.

The different peaks corresponding to the toxins are not easily identifiable by non-expert HPLC readers, so the figure is not essential. Could you also include this figure in the supplementary materials?

References:

The reference No. 16 and No. 17 are not reported in the text. Delete or insert them in the text.

Check the reference No. 14

Please, can you pay attention to the numbering of the references in the text because they are not always in order of numbering.

Author Response

(The authors gave the same response as above.)

Round 2

Reviewer 1 Report

Hello dear,

By checking the new version of the manuscripts, it seems that appropriate changes have taken place It would have been better if the study philosophy and sampling was designed in such a way that it is not only from a specific country and has the ability to generalize with more reliability.

Best regards,

Author Response

Dear Professor,

We greatly appreciate all the critiques and comments from you. Those comments are extremely helpful for us to improving our paper, and they provide valuable guidance for our future study. According to these comments, we have carefully improved our manuscript.

We have made improvments in the methodology and results sections as recommended, and these revisions made to the manuscript were marked up using the “Track Changes” function to be easily viewed by the editors and reviewers.

Reviewer 2 Report

the authors answered adeguately to all request

  •  

Author Response

Dear Professor,

We greatly appreciate all the critiques and comments raised from you. Those comments were extremely helpful for us to improving our paper, and they provided valuable guidance for our future study. According to these comments, we have carefully improved our manuscript,

Yours sincerely,

Associ. Prof.  Ahmed M. Ismail
